# Factors Associated with Severe COVID-19 and Post-Acute COVID-19 Syndrome in a Cohort of People Living with HIV on Antiretroviral Treatment and with Undetectable HIV RNA

**DOI:** 10.3390/v14030493

**Published:** 2022-02-28

**Authors:** Maria Mazzitelli, Mattia Trunfio, Lolita Sasset, Davide Leoni, Eleonora Castelli, Sara Lo Menzo, Samuele Gardin, Cristina Putaggio, Monica Brundu, Pietro Garzotto, Anna Maria Cattelan

**Affiliations:** 1Infectious and Tropical Diseases Unit, University Hospital of Padua, 35128 Padua, Italy; lolita.sasset@aopd.veneto.it (L.S.); davide.leoni@aopd.veneto.it (D.L.); eleonora.castelli@aopd.veneto.it (E.C.); sara.lomenzo@aopd.veneto.it (S.L.M.); samuele.gardin@aopd.veneto.it (S.G.); cristina.putaggio@aopd.veneto.it (C.P.); monicabrundu92@gmail.com (M.B.); pietro.garzotto@gmail.com (P.G.); annamaria.cattelan@aopd.veneto.it (A.M.C.); 2Infectious Disease Unit, Department of Medical Sciences, University of Turin at Amedeo di Savoia Hospital, 10149 Turin, Italy; mattia.trunfio@edu.unito.it

**Keywords:** SARS-CoV-2, COVID-19, PACS, post-acute COVID-19 syndrome, HIV, AIDS, people living with HIV, PLWH

## Abstract

SARS-CoV-2 can produce both severe clinical conditions and long-term sequelae, but data describing post-acute COVID-19 syndrome (PACS) are lacking for people living with HIV (PLWH). We aimed at assessing the prevalence and factors associated with severe COVID-19 and PACS in our cohort. We included all unvaccinated adult PLWH on antiretroviral treatment and plasma HIV-RNA < 40 cp/mL since at least six months before SARS-CoV-2 infection at the Infectious and Tropical Diseases Unit of Padua (Italy), from 20 February 2020 to 31 March 2021. COVID-19 severity was defined by WHO criteria; PACS was defined as the persistence of symptoms or development of sequelae beyond four weeks from SARS-CoV-2 infection. Demographic and clinical variables were collected, and data were analyzed by non-parametric tests. 123 subjects meeting the inclusion criteria among 1800 (6.8%) PLWH in care at the Infectious and Tropical diseases Unit in Padua were diagnosed with SARS-CoV-2 infection/COVID-19 during the study period. The median age was 51 years (40–58), 79.7% were males, and 77.2% of Caucasian ethnicity. The median CD4+ T-cell count and length of HIV infection were 560 cells/mmc (444–780) and 11 years, respectively. Of the patients, 35.0% had asymptomatic SARS-CoV-2 infection, 48% developed mild COVID-19, 17.1% presented moderate or severe COVID-19 requiring hospitalization and 4.1% died. Polypharmacy was the single independent factor associated with severe COVID-19. As for PACS, among 75 patients who survived SARS-CoV-2 symptomatic infection, 20 (26.7%) reported PACS at a median follow-up of six months: asthenia (80.0%), shortness of breath (50.0%) and recurrent headache (25.0%) were the three most common complaints. Only the severity of the COVID-19 episode predicted PACS after adjusting for relevant demographic and clinical variables. In our study, PLWH with sustained viral suppression and good immunological response showed that the risk of hospital admission for COVID-19 was low, even though the severity of the disease was associated with high mortality. In addition, the likelihood of developing severe COVID-19 and PACS was mainly driven by similar risk factors to those faced by the general population, such as polypharmacy and the severity of SARS-CoV-2 infection.

## 1. Introduction

Since late December 2019, the Severe Acute Respiratory Syndrome Coronavirus 2 (SARS-CoV-2) pandemic has had an important impact on the approach to several chronic conditions, including HIV infection [1,2,3]; indeed, the pandemic disrupted the normal flow of linkage to care for these patients and lead to a tailored management of SARS-CoV-2 infection in people living with HIV (PLWH). SARS-CoV-2 infection can range from asymptomatic carriage to mild flu-like symptoms, to severe pneumonia and respiratory failure requiring admission to an intensive care unit (ICU) [4,5]. From the beginning of the pandemic, several reports have been published both on the association of HIV with an increased risk of developing symptomatic SARS-CoV-2 infection, and poor prognosis of the 2019 novel Coronavirus Disease (COVID-19) in PLWH [6,7]. By contrast, other authors did not detect HIV as risk factor for more severe COVID-19, reporting risk factors of severe pneumonia and related death similar to those observed for the general population (older age, male gender, comorbidities) [8,9,10,11,12]. In fact, some authors noted that HIV-related lymphopenia can have a protective role in preventing the severe clinical manifestations of COVID-19 [13]. However, the debate is still ongoing, and to-date, the WHO include HIV as an independent risk factor for severe and critical COVID-19, regardless of the varying characteristics of HIV status. In addition, the role of antiretrovirals in the outcome of PLWH with COVID-19 has been controversial and no role has been found for cART in the general population [14,15,16,17,18,19]. Indeed, studies confirmed no efficacy of protease inhibitors in reducing risk of complications nor in improving clinical outcomes [18,19]. In a recent pilot study, exposure to tenofovir/emtricitabine appeared to accelerate viral clearance in non-HIV people, but data are not conclusive [20].

Even fewer data are available on the incidence, prevalence, and characteristics of post-acute COVID-19 syndrome (PACS) among PLWH. Currently, the pathophysiological mechanisms contributing to PACS onset are still poorly understood and this issue is of growing clinical concern in PLWH. There could be an increased risk of PACS in the overall HIV-positive population due to chronic inflammation, immune dysfunction, and alterations in the immunological response against SARS-CoV-2, but this result may be driven by specific subgroups of patients featured by uncontrolled viral replication or severe immunological deficit. Therefore, the aims of this work were: (i) to assess factors associated with severe COVID-19 among all PLWH with undetectable HIV viraemia at our Clinic and (ii) to assess the prevalence and risk factors associated with PACS in the same population. 

## 2. Materials and Methods

This is a prospective observational cohort study including all adult PLWH linked to care at our centre (Infectious and Tropical Diseases Unit, Padova, Italy), not vaccinated for SARS-CoV-2, with plasma HIV-RNA < 40 cp/mL for at least six months, and who were diagnosed with SARS-CoV-2 infection (either by RT-PCR on nasopharyngeal swab and/or serology) from 20 February 2020 to 31 March 2021. SARS-CoV-2 diagnosis was made through RT-PCR on nasal-pharyngeal swab performed both for the presence of symptoms and to every access to our hospital during the study period; similarly, SARS-CoV-2 serology was performed systematically at every access to hospital for routine clinical checks. The study protocol was approved by the Ethics Committee (n° study protocol approval 4933/AO/2020), and all patients were required to sign informed consent for participation. The following information were collected for each patient from medical records to detect baseline characteristics at the time of SARS-CoV-2 diagnosis: age, gender, ethnicity, year of HIV diagnosis, AIDS defining illnesses, CD4+ T cell nadir and CD4+ T cell, plasma HIV-RNA (for asymptomatic SARS-CoV-2 serological diagnosis, last available CD4+ T cell count and HIV-RNA were considered), comorbidities (hypertension, diabetes, obesity, HBV and HCV-coinfection, ischemic heart disease, osteoporosis, chronic renal failure, neurological disorders, psychiatric disorders, cirrhosis, cancer, immune disorders, lung diseases), list of co-medications, and antiretroviral treatment. The date of SARS-CoV-2 infection for SARS-CoV-2 PCR-based diagnosis or the date of SARS-CoV-2 serological detection, SARS-CoV-2 related signs and symptoms, SARS-CoV-2 infection clinical phenotype (need of hospitalization, need of oxygen support, recovery/death), and the presence and type of PACS were also recorded. PACS has been defined as the persistence of symptoms and/or delayed or long-term complications beyond four weeks from acute SARS-CoV-2 infection [21]. To detect the presence of PACS, patients were followed-up for six months after the diagnosis of COVID-19 (last follow-up time point for the last patient included 30 September 2021). 

SARS-CoV-2 infection severity was classified as follows: asymptomatic infections, COVID-19 cases not requiring hospital admission, and COVID-19 cases requiring hospital admission due to signs/symptoms related to SARS-CoV-2 infection (moderate/severe COVID-19). Asymptomatic and/or mild symptomatic patients who tested positive at nasopharyngeal swabs were followed-up by telemedicine and strictly monitored for the need of oxygen support and hospitalization. Patients requiring oxygen support but not admitted to hospital due to bed shortage were treated at home and monitored with daily telemedicine visits and were considered among PLWH with moderate/severe COVID-19. 

To further assess the potential effects on COVID-19 severity and PACS development by variables that we did not collect, such as clinical management changes during the first year of pandemic, accessibility to care, or the emergence of SARS-CoV-2 variants during the entire study period, we divided the study population according to tertiles of the entire study period, knowing that: analgesic and antipyretics were administered only to not hospitalized patients for the entire period, while patients admitted to hospital were administered with corticosteroids plus remdesivir and heparins during the second and last period tertile (as per internal hospital protocol, no immunomodulant agents as tocilizumab were given to PLWH at that time), while treatments were less homogeneous during the first tertile; bed availability and access to care was at its best during the second tertile when the first wave flawed and before the beginning of the second wave; SARS-CoV-2 alpha variant of concern may have represented part of the cases detected in the third period tertile only. Monoclonal antibodies became available in the outpatient’s setting during the final tertile.

### Statistical Analysis

All data were collected in an electronic spreadsheet. Data were reported as median (interquartile range, IQR) for continuous variables and absolute number (percentage) for categorical variables, as appropriate. Non-parametric tests were used for the analysis (Mann–Whitney U test, Kruskal–Wallis test, chi-squared, and Fisher’s exact test). The odds ratio (OR) and 95% confidence interval (95% CI) were calculated as effect measure for risk association. Multivariate multinomial and binary logistic regression (standard entry method) were run including variables of significance at univariable analysis (level of significance was set as *p*-value < 0.05). Data were analysed through SPSS v27 (IBM Stat. Corp., Armonk, NY, USA). 

## 3. Results

### 3.1. Study Population

One hundred and twenty-four patients belonging to our cohort of 1800 PLWH that regularly attend our HIV Clinic were diagnosed with SARS-CoV-2 infection (period prevalence of 6.9%). Among them, one subject was not on cART and had a concurrent AIDS diagnosis while developing COVID-19 and was excluded from the analyses. Among the remaining 123 patients, 15 (12.2%) were detected by serology (during their routine clinical check) and 108 (87.8%) by nasopharyngeal swabs. Among those who were detected by PCR on nasopharyngeal swab, 103 were detected during the routine clinical check, and 5 at the emergency department for the onset of respiratory failure. According to the study period, none of the study patients received SARS-CoV-2 vaccination. Overall, 98/123 individuals (79.7%) were male, 95 (77.2%) were of Caucasian ethnicity, median age and current CD4 count were 51 years (40–58) and 560 cells/mmc (444–780), respectively. Median length of HIV disease was 12 years (5–19). Twenty-four (19.5%) had at least one AIDS episode in their past medical history. All PLWH had plasma HIV-RNA <40 cp/mL, as per inclusion criteria. As for comorbidities, the most common were hypertension (38, 30.9%), diabetes (22, 17.9%), and overweight/obesity (19, 15.4%). Thirty-nine (31.7%) PLWH were active smokers. Twenty-seven (21.9%) and 36 (29.3%) had anti-HCV and anti-HBc antibodies, but none had replicating viruses. Nineteen individuals (15.4%) meet the criteria for polypharmacy (more than two non-antiretroviral drug per day). 

Among the 123 SARS-CoV-2 positivity, 43 (35.0%) resulted in asymptomatic infections, 59 developed mild COVID-19 not requiring hospitalization (48.0%), and 21 (17.1%) had a moderate/severe COVID-19 that required hospitalization and oxygen support (8 patients were provided with oxygen support at home and monitored daily by territorial services due to the lack of hospital beds during the zenith of the first pandemic wave and were counted among moderate/severe cases). Five patients (4.1%) among the 21 hospitalized PLWH required intensive care unit admission and eventually died. Tab.1 reports the demographic, clinical and viro-immunological features of the study population stratified by SARS-CoV-2 infection severity. 

### 3.2. Prevalence and Factors Associated with Moderate/Severe COVID-19

Overall, 21/123 (17.1%) PLWH developed moderate/severe COVID-19 and all five deaths occurred in this group. Table 1 reports the comparison across patients grouped by clinical severity. Overall, the three groups differed in terms of age, ethnicity, prevalence of osteoporosis, psychiatric disorders, polypharmacy, and previous AIDS episodes as well as for length of HIV infection. Indeed, asymptomatic cases had most of the SARS-CoV-2 infections detected by a posteriori serology in the study population (Table 1).

At multivariate analysis, after including univariate significant variables (age, ethnicity, osteoporosis, polypharmacy, length of HIV infection and previous AIDS episodes), only polypharmacy was independently associated with an increased risk of more severe SARS-CoV-2 infections (aOR 9.4 [1.4–64.7], *p* = 0.023 for mild COVID-19 versus asymptomatic infections and aOR 12.9 [1.8–94.4], *p* = 0.011 for moderate/severe COVID-19 versus COVID-19 not requiring hospitalization).

### 3.3. Prevalence and Factors Associated with PACS

Among the 75 PLWH who experienced symptomatic SARS-CoV-2 infection and survived, 20 patients (26.7%) reported PACS at a median follow up time of six months. The three most common reported issues were: asthenia (16, 80.0%), efforts-related shortness of breath (10, 50.0%) and recurrent headache (5, 25.0%). Two patients who had mild COVID-19 reported important cardiovascular complications: one was diagnosed with myocardial infarction four weeks after COVID-19 resolution (this patient immediately after healing from COVID-19 continued experiencing shortness of breath and chest pain) and one developed persistent heart rhythm alterations. Significant hair loss with a relevant impact on the quality of life was reported in three patients (15.0%); the loss of smell remained in two patients (10.0%), while one patient had chronic diarrhoea. Compared to patients with no COVID-19 related sequelae, patients who eventually developed PACS did not differ in terms of demographic, clinical, nor HIV-related viro-immunological parameters but for larger prevalence of polypharmacy, psychiatric disorders, previous AIDS episodes, and more severe COVID-19 (Table 2). 

At multivariate analysis, that included severity of COVID-19 (not requiring versus requiring hospitalization), psychiatric disorders, polypharmacy, and previous AIDS episodes, only the severity of the COVID-19 episode independently associated with PACS development (aOR 6.61 [1.74–25.20], *p* = 0.006).

## 4. Discussion

In our cohort of PLWH, we detected an overall prevalence of SARS-CoV-2 infection and PACS of 6.9% and 26.7%, respectively. In PLWH, the overall prevalence of COVID-19 seems to be similar to that reported in the general population [22], however PLWH seem to be at higher risk of more severe COVID-19 and deaths from SARS-CoV-2 infection [23,24]. In our study, we observed an overall prevalence of SARS-CoV-2 infections and severe COVID-19 of 6.9% (123/1800) and 17.1% (21/123), respectively. The data on the overall prevalence, even in our region, did not significantly differ between PLWH and the general population of Italy (6.9% vs. 6%) [25], while the prevalence of severe COVID-19 seemed to be significantly higher (17.1% vs. 4.4%) [25]. This seems to confirm, as previously reported, that HIV itself can be a risk factor for the acquisition of SARS-CoV-2 infection [22,26]. The prevalence of complications (i.e., requiring hospitalization) from COVID-19 in PLWH ranges from 25% to 68% [27,28]. In our cohort, most patients (82.9%) had asymptomatic to moderate disease, not requiring hospitalization. Our hospitalization rate was lower than that reported from other experiences [29], probably due to different approaches to the disease. Indeed, in our experience, we treated patients early at home as soon as they were diagnosed with SARS-CoV-2 infection, and admission had been reserved only to moderate/severe cases. Patients were also monitored by telemedicine services by both an infectious disease specialist and a general practitioner. In two PLWH who had several risk factors for severe COVID-19, we used treatment with bamlanivimab/etesevimab, with a good clinical outcome. Moreover, our mortality rate was 3.6%, the lowest currently described [30]. 

At the multivariable model performed among all cases, we observed that polypharmacy was the only factor significantly associated with an increased risk of developing more severe SARS-CoV-2 infection. Polypharmacy can be considered a proxy of multimorbidity, even more reliable than multimorbidity alone [31,32]. Multimorbidity in PLWH, due to ageing and effective antiretrovirals, is an important clinical issue and is more represented in PLWH compared to the general population at the same age [33]. At the univariate analysis, age and ethnicity, as well as in the general population, were also recognised as risk factors for severe COVID-19 [34,35,36]. Unfortunately, the association was not confirmed in the multivariable analysis. 

At univariate analysis, severe COVID-19 associated with factors that were identified both in the general population (psychiatric disorders) and HIV-specific factors (length of HIV infection), as previously observed by other authors [24,37]. The links between psychiatric disorders and more severe COVID-19 disease in this population deserves to be investigated further, but a possible effect of comedications that may lead to depression of respiratory driver has been already observed in the general population [38]. However, to date, no data are available to support this observation and we did not analyse in detail each single anti-psychiatric agent patients were taking. Another possible explanation for this association is that patients with psychiatric comorbidities may pay no attention to relevant symptoms, underestimate the risk of disease, and therefore remain undertreated or present to medical observation late [39]. We can also speculate that mental health issues are frequent in people with COVID-19 and therefore there could be a reversal causality between psychiatric disorders and COVID-19.

As for HIV-specific factors, the length of HIV was significantly associated with the risk of severe COVID-19 at univariate, but not confirmed at multivariate analysis; it is reasonable to think that PLWH with a longer history of disease, even if with suppressed HIV viral load and a good CD4+ T-cell count, may have higher levels of inflammation and immune dysfunction [40,41] that may lead patients to develop more severe COVID-19. In our cohorts, PLWH who died were all diagnosed with COVID-19 in the emergency department. They had a duration of HIV twice as long as the asymptomatic or moderately infected cases, and they were all late presenters at HIV diagnosis (i.e., CD4 T cell count < 350 cells/mmc) [42]. Moreover, they came to observation in the emergency department with severe SARS-CoV-2 pneumonia and during intensive care unit admission developed multi-drug resistant bacterial infections as well as fungal infections. 

An intriguing association with the risk of severe COVID-19, but detected only in the univariate analysis, was the presence of previous AIDS pulmonary infections. Interestingly, this association was also detected with other viral conditions, such as influenza [43,44,45,46], and may be due to a “more fragile” lung. Therefore, even if this finding needs to be confirmed on a greater number of patients, since past pulmonary AIDS events may be a driver of severe COVID-19, both prevention strategies (i.e., vaccines) and early treatment (i.e., early hospitalizations and use of monoclonal antibodies) should be promptly implemented in PLWH with previous episodes of AIDS events involving lungs (i.e., *Pneumocystis jirovecii* pneumonia). As for antiretrovirals, we did not detect any difference in COVID-19 course between those who were on boosted protease inhibitors or non-nucleoside agents and other antiretroviral regimes.

As for symptoms at presentation, interestingly, fever was present in 74.6% and 95.2% PLWH who did not require and who required hospitalization, respectively. This prevalence seems to be in contrast with data coming from other immunocompromised populations, such as those with cancer, where fever prevalence was found to be 59% [47].

As for PACS, its prevalence reaches up to 32.6% in the general population, but data about its real prevalence and whether PLWH are more likely to develop PACS is still unknown [21,48]. To the best of our knowledge, this is the first work assessing the prevalence of and factors associated with PACS in PLWH. One out of four patients with symptomatic SARS-CoV-2 infection reported PACS in our cohort. The number of symptoms and the severity of COVID-19 episode were associated with PACS in our cohort as it was already observed in the general population [21,48]. Notably, two PLWH developed cardiovascular complications (myocardial infarction and arrythmia) during the three months of follow-up. Whether these conditions were caused by COVID-19 or were just a coincidence need to be clarified. Certainly, two main hypotheses should be considered. Firstly, it is well known how viral infections and pneumonia may cause and worsen cardiovascular events, especially in patients who are at increased risk [49,50]. Secondly, in patients with known pre-existing cardiovascular risk factors, a cytokine storm induced by the presence of SARS-CoV-2 may have worsened and accelerated the cardiac damage and caused stress cardiomyopathy [51,52]. Moreover, the effect of interactions between HIV and SARS-CoV-2 on chronic inflammation is unknown. Further studies with a different design are necessary to strengthen these speculations. Furthermore, some symptoms, such as asthenia and arthralgia, are more likely to be associated with PACS and to persist over time. Moreover, nor demographic, clinical, or HIV-related viro-immunological parameters seemed to have an impact on the presence of PACS, except polypharmacy, previous AIDS episodes, and more severe COVID-19. However, surveillance for PACS should be performed in PLWH as well as in the general population since it may significantly affect quality of life. Our experience showed the importance of monitoring for PACS, especially for cardiovascular complications, even if COVID-19 infection was mild. 

Indeed, our study presents several limitations that should be acknowledged: its observational nature, the lack of a control group (both without HIV and with HIV, but with not suppressed HIV-RNA), and the low number of PLWH included. As such, due to the small sample size and the observational nature of the study, we were not able to properly assess the possible impact on clinical course and on PACS of both viral variants over time and the different treatments which became available over the study period. As a further limitation, we cannot avoid mentioning the possible recalling bias from patients who were identified by serology. As for the last, due to the nature of our study design, we could not calculate an a priori sample size to statistically support the power of some of our analyses and our regression models’ results are limited by the low number of detected events in the entire sample, as highlighted by the large confidence intervals of the odds ratios reported. Future studies are required to confirm our findings and to provide further evidence about the prevalence, characteristics and factors associated with PACS in PLWH. 

## Figures and Tables

**Table 1 viruses-14-00493-t001:** Comparison of baseline demographic, clinical, and HIV-related parameters grouped according to COVID-19 severity.

Variable	Asymptomatic Infections (*n* = 43)	Symptomatic Infections Not Requiring Hospital Admission (*n* = 59)	Symptomatic Infections Requiring Hospital Admission (*n* = 21)	*p*
Age, years, median (IQR)	47 (40–55)	51 (40–58)	58 (41–65)	0.007
Male sex, *n* (%)	32 (74.4%)	49 (83.1%)	17 (80.9%)	0.56
Caucasian, *n* (%)	27 (62.8%)	51 (86.4%)	17 (80.9%)	0.018
BMI, median (IQR)	25.8 (23.0–27.0)	25.0 (23.1–27.7)	26.0 (24.4–28.7)	0.251
Number of comorbidities, median (IQR)	1 (0–2)	1 (0–2)	2 (0–4)	0.473
Current smoker, *n* (%)	13 (30.2%)	17 (28.8%)	9 (42.9%)	0.481
HIV infection, median years (IQR)	10 (4–16)	10 (5–17)	21 (12–27)	0.004
Current CD4 count, median cells/mmc (IQR)	560 (440–777)	558 (469–780)	637 (281–836)	0.961
CD4/CD8 ratio, median (IQR)	0.88 (0.56–1.12)	0.87 (0.64–1.10)	0.73 (0.30–1.1)	0.961
CD4 count nadir, median cells/mmc (IQR)	367 (227–563)	368 (260–591)	203 (48–551)	0.313
Previous AIDS episode, *n* (%)				
Any	6 (13.9%)	9 (15.2%)	9 (42.9%)	0.013
Pulmonary	1 (2.3%)	3 (5.1%)	6 (28.6%)	0.001
Type of cART, *n* (%)				
Mono-dual therapy	11 (25.6%)	18 (30.5%)	3 (14.3%)	0.421
PI-based	3 (6.9%)	5 (8.5%)	4 (19.0%)	
INI-based	13 (30.2%)	14 (23.7%)	8 (38.1%)	
NN-based	16 (37.2%)	22 (37.3%)	6 (28.6%)	
Positive anti-HCV Ab, *n* (%)	11 (25.6%)	9 (15.2%)	7 (33.3%)	0.18
Positive anti-HBc Ab, *n* (%)	13 (30.2%)	20 (33.9%)	3 (14.3%)	0.236
Comorbidity, *n* (%)				
Obesity	7 (16.3%)	8 (13.5%)	4 (19.0%)	0.823
Hypertension	11 (25.6%)	16 (27.1%)	11 (52.4%)	0.065
Diabetes	4 (9.3%)	13 (22.0%)	5 (23.8%)	0.19
Ischaemic hearth disease	1 (2.3%)	4 (6.8%)	3 (14.3%)	0.191
Osteoporosis	8 (18.6%)	2 (3.4%)	4 (19.0%)	0.028
Chronic renal failure	2 (4.6%)	4 (6.8%)	1 (4.8%)	0.883
Neurological disorders	4 (9.3%)	7 (11.9%)	3 (14.3%)	0.831
Psychiatric disorders	6 (13.9%)	3 (5.1%)	5 (23.8%)	0.056
Cirrhosis	3 (6.9%)	0 (0.0%)	1 (4.8%)	0.135
Cancer	2 (4.6%)	4 (6.8%)	3 (14.3%)	0.375
Immune disorders	1 (2.3%)	2 (3.4%)	1 (4.8%)	0.873
Lung disorders	1 (2.3%)	3 (5.1%)	2 (9.5%)	0.455
Polypharmacy, *n* (%)	2 (4.6%)	8 (13.5%)	9 (42.9%)	<0.0005
Number of non-antiretroviral drugs, median (IQR)	1 (0–2)	1 (0–2)	4 (1–5)	0.111
SARS-CoV-2 diagnosis, *n* (%)				
PCR-based	33 (76.7%)	54 (91.5%)	21 (100%)	<0.0005
Serological	10 (23.3%)	5 (8.5%)	0 (0.0%)	
Calendar date of SARS-CoV-2 infection, *n* (%) *				
March–mid July 2020	3/33 (9.1%)	0/54 (0%)	3/21 (14.3%)	0.329
Mid July–November 2020	4/33 (12.1%)	6/54 (11.1%)	1/21 (4.8%)	
December 2020–March 2021	26/33 (78.8%)	48/54 (88.9%)	17/21 (80.9%)	
Number of signs and symptoms, median (IQR)	0 (0–0)	3 (2–4)	4 (4–5)	NA
Signs and symptoms, *n* (%)				
Pharyngitis	0 (0%)	13 (22.0%)	4 (19.0%)	NA
Olfactory dysfunction		19 (32.2%)	5 (23.8%)	
Gustatory dysfunction		19 (32.2%)	6 (28.6%)	
Fever		44 (74.6%)	20 (95.2%)	
Cough		25 (42.4%)	16 (76.2%)	
Asthenia		29 (49.1%)	11 (52.4%)	
Dyspnoea		6 (10.2%)	15 (71.4%)	
Diarrhoea		6 (10.2%)	2 (9.5%)	
Arthromyalgia		15 (25.4%)	5 (23.8%)	
Headache		8 (13.6%)	6 (28.6%)	
COVID-19-related death, *n* (%)	0 (0%)	0 (0.0%)	5 (23.8%)	NA
PACS, *n* (%)	0 (0%)	10 (16.9%)	10/16 (62.5%)	NA

Legend: IQR, interquartile range; BMI, body mass index; cART, combination antiretroviral therapy; PI, protease inhibitors; INI, integrase inhibitors; NN, non-nucleoside reverse transcriptase inhibitors; HCV, hepatitis C virus; HBV, hepatitis B virus; PACS, post-acute COVID-19 syndrome; NA, not assessed. * Among PCR-tested PLWH only (*n* = 108). Kruskal–Wallis test and non-parametric t distribution test as for proper variable type were performed between the three groups.

**Table 2 viruses-14-00493-t002:** Comparison of baseline demographic, clinical, and HIV-related parameters between PLWH with and without post-acute COVID-19 syndrome.

Variable	Post-COVID-19 Syndrome (*n* = 20)	No COVID-19-Related Sequelae (*n* = 55)	*p*
Age, years, median (IQR)	53 (40–58)	53 (40–58)	0.93
Male sex, *n* (%)	18 (90.0%)	43 (78.2%)	0.249
Caucasian, *n* (%)	16 (80.0%)	47 (85.4%)	0.571
BMI, median (IQR)	25.6 (23.6–28.0)	25.3 (23.1–28.4)	0.741
Number of comorbidities, median (IQR)	1 (0–4)	1 (0–2)	0.9
Current smoker, *n* (%)	7 (35.0%)	16 (29.1%)	0.626
HIV lenght, median years (IQR)	12 (7–23)	11 (5–20)	0.638
Current CD4 count, median cells/mmc (IQR)	580 (345–770)	558 (450–786)	0.848
CD4/CD8 ratio, median (IQR)	0.82 (0.60–1.14)	0.87 (0.64–1.01)	0.958
CD4 count nadir, median cells/mmc (IQR)	271 (45–595)	388 (261–607)	0.26
Previous AIDS episode, *n* (%)	8 (40.0%)	9 (16.4%)	0.032
Type of cART, *n* (%)			
Mono-dual therapy	3 (15.0%)	17 (30.9%)	0.172
PI-based	1 (5.0%)	7 (12.7%)	
INI-based	6 (30.0%)	14 (25.4%)	
NN-based	10 (50.0%)	17 (30.9%)	
Positive anti-HCV Ab, *n* (%)	6 (30.0%)	9 (16.4%)	0.195
Positive anti-HBc Ab, *n* (%)	3 (15.0%)	20 (36.4%)	0.078
COVID-19 requiring hospitalization, *n* (%)	10 (50.0%)	6 (10.9%)	<0.0005
Calendar date of SARS-CoV-2 infection, *n* (%) *			
March–mid July 2020	1/20 (5.0%)	1/50 (2.0%)	0.258
Mid July–November 2020	3/20 (15.0%)	4/50 (8.0%)	
December 2020–March 2021	16/20 (80.0%)	45/50 (90.0%)	
Comorbidity, *n* (%)			
Obesity	2 (10.0%)	10 (18.2%)	0.396
Hypertension	6 (30.0%)	16 (29.1%)	0.939
Diabetes	3 (15.0%)	13 (23.6%)	0.423
Ischaemic hearth disease	1 (5.0%)	4 (7.3%)	0.729
Osteoporosis	3 (15.0%)	2 (3.6%)	0.083
Chronic renal failure	1 (5.0%)	4 (10.9%)	0.433
Neurological disorders	3 (15.0%)	5 (9.1%)	0.729
Psychiatric disorders	5 (25.0%)	1 (1.8%)	0.001
Cirrhosis	1 (5.0%)	0 (0.0%)	0.097
Cancer	2 (10.0%)	4 (7.3%)	0.702
Immune disorders	0 (0.0%)	3 (5.4%)	0.29
Lung disorders	3 (15.0%)	2 (3.6%)	0.083
Polypharmacy, *n* (%)	7 (35.0%)	7 (12.7%)	0.03
Number of non-antiretroviral drugs, median (IQR)	1 (1–5)	1 (0–2)	0.79

Legend: IQR, interquartile range; BMI, body mass index; cART, combination antiretroviral therapy; PI, protease inhibitors; INI, integrase inhibitors; NN, non-nucleoside reverse transcriptase inhibitors; HCV, hepatitis C virus; HBV, hepatitis B virus; PACS, post-acute COVID-19 syndrome. * Among PCR-tested PLWH only (*n* = 70). Mann–Whitney U test and non-parametric t distribution test as for proper variable type were performed.

## Data Availability

Not applicable.

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
