# Peer review of "Factors Associated with Severe COVID-19 and Post-Acute COVID-19 Syndrome in a Cohort of People Living with HIV on Antiretroviral Treatment and with Undetectable HIV RNA"

_viruses, 2022, doi:10.3390/v14030493_

Round 1
Reviewer 1 Report
This manuscript examines the prevalence and factors associated with severe COVID-19 and post-acute COVID-19 syndrome (PACS) among PLWH in the cohort of patients at the Infectious and Tropical Diseases Unit, Padua, Italy. This is an important study, and although a number of investigations have been published on this topic, all additions to this collection from different setting are important to contribute. The authors identify some factors associated with asymptomatic or symptomatic CoVID-19 diagnosis, and factors associated with developing PACS, which appears to be related to more severe SARS-CoV-2 disease. These descriptive aspects of the paper are important.
A strength of the study is that it is based within a large cohort of PLWH and that the authors also retrospectively tested individuals for SARS-CoV-2 serology at their routine clinic visits. A large number of variables were extracted from clinical records relating to demographics, immunological and virological parameters, comorbidities and treatments and are reported in the paper. A major limitation that this is a very small study, and therefore has little power to analyse associations by statistical methods. A few associations are highlighted, but because of the limited power these can only serve as pointers for hypothesis generation for further investigations. In addition, the manuscript also requires careful checking for consistency and English language editing for spellings, punctuation and grammar. A few examples are listed below, but more extensive language editing is required.
Abstract
- Instead of “since the beginning of the epidemic” (line 21), please add precise dates. Similarly precise dates of the study should be given in the main Methods section (Line 83)
- The sentence “Almost 50% of patients had asymptomatic SARS-CoV-2 infection, 24.5% presented moderate or severe COVID-19, 9.4% were hospitalised and 3.6% died.” (lines 27-29) does not agree with the Results which report 43/123 (35.0%) of asymptomatic SARS-CoV-2 infections, and 59/123 (48%) and 21/123 (17.1%) individuals with moderate (non-hospitalised) and severe (hospitalised) CoVID-19, respectively, as well as 5 deaths (5/123 = 4.1%). Please clarify the denominators
- In line 30-32, PACS symptoms are described for 102 individuals even though (line 30) only 20/75 individuals with symptomatic CoVID-19 were included in the analysis? Again this is confusing.
Methods
- Line 79 describes the study as a prospective observational cohort study, but in the discussion (line 302) it is called retrospective. Please clarify – is this a retrospective analysis of CoVID patients nested within a prospective HIV cohort study?
- Line 83 – please provide precise dates (with numbered days) for the start and end of the study period, and the end of follow-up for the PACS study. It would also be informative to report (and list in the tables) the calendar times of the SARS-CoV-2 infections (e.g. whether they occurred during the first wave in March-April 2020, or the wave later in the year (November 2020 to February 2021 when the Alpha variant was beginning to circulate) as the treatments available and access to care would likely differ between these waves, and which may influence the occurrence of PACS.
Results
Section 3.1, study population:
- Case ascertainment: Have all 1800 PLWH in the cohort been screened for SARS-CoV-2 serology and attended at the clinic, or could there be some selection bias due to individuals missing follow-up, or attending for CoVID-19 treatment at a different site? This impacts on prevalence calculations.
- Line 134: detected by nasopharyngeal swabs – this is by PCR, which should be stated.
- Lines 134-136: “108 (87.8%) of SARS-CoV-2 infections were detected by nasopharyngeal swabs”. In the next sentence “Among those who were detected by nasopharyngeal swab 105 were detected during the routine clinical check, while 5 at the emergency department for the onset of respiratory failure”. Note: 105+5 = 110, so this is more than the 108 mentioned in line 134.
- Line 140: Clarify where brackets quote IQRs
- Line 147: 43/123 is 35.0% (rounding error)
- Line 152: How do you define “a really severe CoVID” - I suggest deleting “a really”
- Table 1 shows way too many P values and presents the comparisons in a very confusing manner. Given the small number of study participants, it would be better to limit the comparisons to the three main groups in the table (shown with P***) and remove the comparison shown in the column on the right (P *). It is not clear whether this latter analysis compares the moderate and severe cases as listed on the table, or whether it refers to a different grouping based on hospitalisation. Remember that with multiple testing it is possible that several comparisons (around 1 in 20) can return P values <0.05 purely by chance. The P*** values should be reported in a column on the right hand side, and please don’t highlight or italicise some P values e.g. those considered significant at 5% significance level.
- Table 1
- Title: Please clarify whether these are baseline data, and that baseline is defined as the date at SARS-CoV-2 diagnosis or of the serology test.
- Any previous AIDS episode – there is a missing bracket ( “(“ ) for one of the percentages
- Ischemic heart disease (not hearth disease)
- For the * footnote, “P-values only for symptomatic patients requiring (n=10) vs not requiring hospitalization (n=71)” – but the table shows data for 80 symptomatic patients, divided according to severity. I assume these P-values refer to a different grouping to that shown in the table? In this case they should definitely be removed from the table.
- The footnote could also show abbreviations for ART classes, HCV, HBV and PACS
Section 3.2, Prevalence and factors associated with a moderate/severe COVID-19
- Altogether 21/123 individuals had severe CoVID-19, which is perhaps better described as a proportion, as prevalence “in our cohort” is ambiguous (vs. the cohort of PLWH, or the analysis of SARS-CoV-2 positive individuals)
- A detailed description of the factors shown in Table 1 is warranted, but this should be based on the overall (3-group comparison) and P values in Table 1 (P***).
- Given the small numbers and the lack of power, I am not sure multivariable analysis is warranted. Roughly, according to the “rule of 10” (which suggests that one predictive variable can be studied for every ten events) with the smallest group having just 21 events, only two variables may be adjusted for. I would also be weary of using automated forward selection with these sparse events to avoid collinearity or unintentionally adjusting for mediators or colliders.
- For any factors that appear to be significantly associated with the outcome of severe CoVID (or in the next section, PACS), consider power, and whether the result might be due to chance, bias or confounding. For proposed causal associations, there should be some plausible mechanism.
Section 3.3, Prevalence and factors associated with PACS
- As above, due to lack of power this can only be descriptive. In this context, a discussion of when (Calendar time) the individuals who developed PACS were first diagnosed with CoVID (as calendar time) and their treatments at the time may be relevant. Again, multivariable modelling is probably not justified.
- Notes about Table 2
- Ischemic heart disease (not hearth disease)
- Missing P value for lung disorders
- There is no footnote, please add.
Discussion.
- As per convention, it would be helpful to briefly present the results from this analysis before discussing them.
- Line 219-220 “In our cohort most patients (90.9%) had asymptomatic to moderate disease, not requiring hospitalization” – where does the 90.9% figure come from? According to Table 1, 21/123 = 17.1% of patients required hospitalisation and presumably 82.9% did not.
- Lines 215-217 comparing PLWH with the general population: the first bracket (6.9% vs. 6%) lists the proportion in PLWH first and then that in the general population, while the second (4,4% vs 17%) shows the general population proportion first and that for PLWH second. Please make consistent.
- Line 218-219 – how do you define a ‘complicated course of CoVID’?
- Line 226-227 describes mAb treatment for 2 individuals. A fuller discussion of how patients were treated during the study and other treatment experiences would be helpful here, especially if it might point to reasons why the study showed that the “mortality rate was 3.6%, the lowest currently described” (line 227-228)
- While mental health issues among PLWH and specifically those who experienced CoVID-19 are important, this study does not have the power to comment on associations with psychiatric disorders or depression, and indeed there could be reverse causality here.
- Line 284 – this should read ‘two main hypotheses should be considered’.
- There are important limitations in this study including the observational nature of the study and low numbers of participants and therefore power. The role of chance and possible biases including recall bias from patients identified from serology or bias due to different treatments experienced by the patients also need to be discussed.
Some minor points of language editing:
- People First Language: As per community request, avoid referring to PLWH as HIV-positive (e.g. line 71, line 201) or describe their HIV infection (line 34, line 161, 166 etc) or call them ‘subjects’. ‘Patient’ or ‘infected’ should only be used in connection with CoVID-19 or SARS-CoV-2.
- Line 31: headsache in 25%, hairloss in 15%, shortness of breat h in 10%, andsevere: headache, hair-loss, breath, and severe
- Line 35: Need to be further investigated
- Line 66: emtricitabile should be emtricitabine
Author Response
Dear Reviewer,
Thanks for your comments which were useful to improve the quality od our work, please find attached the document with our responses.
Kind regards,
Maria Mazzitelli

Reviewer 2 Report
Overall, authors present a comprehensive retrospective study of PLWH on antiretrovirals and with undetectable HIV RNA to determine factors leading to incidence of post-acute Covid-19 syndrome. While their cohort was respectable, it was still limited based upon small number who actually got covid and had PACS.
While study design is sound, there is little new results from this manuscript that will provide insight for clinical practice. Results confirm previous reports that not much difference in COVID-19 severity between PLWH and non-HIV infected individuals, although control not included.
Authors are advised to have an editor review the manuscript for spelling and english grammar to improve quality.
Author Response
Dear Reviewer,
Thanks for your comments which were useful to improve the quality of our work, please find attached the document with our responses.
Kind regards,
Maria Mazzitelli
